# Reconstructing albedo from mean cloud properties

Izabela Wojciechowska<sup>1</sup>, Edward Gryspeerdt<sup>2</sup>

<sup>1</sup>Faculty of Geography and Regional Studies, University of Warsaw, Krakowskie Przedmiescie 30, 00-927 Warsaw, Poland <sup>2</sup>Department of Physics, Imperial College London, London, SW7 2BX, UK

Correspondence to: Izabela Wojciechowska (i.wojciechows2@uw.edu.pl)

**Abstract.** Liquid marine clouds exert a substantial control on the Earth-atmosphere energy system through their large global coverage and high reflectivity of shortwave radiation, resulting in overall negative radiative impact. Previous studies showed that the two dominant factors determining their albedo are cloud fraction (CF) and liquid water path (LWP), but this relationship varies in regions of high aerosol loading. In this work, a simplified kernel was built to assess how well the top of atmosphere (TOA) all-sky albedo ( $\alpha$ ) can be estimated from the given properties of marine liquid clouds: CF, LWP and cloud droplet number concentration ( $N_d$ ), and to what extent this approach applies globally. The study uses data retrieved from MODIS and CERES instruments for a near-global ocean domain (60°S–60°N) covering the period 2003–2021. The results showed that the albedo is only reconstructed to within 10% in less than 40% of cases. Several modifications of investigated method were tested for the improvement in albedo reconstructions. It was found that the number of biases decreases when the maximum solar zenith angle is considered, as well as if the CF–LWP– $N_d$ – $\alpha$  kernel is calculated on a 1° latitude-longitude grid. The findings show that the relationship between the TOA albedo of a scene of clouds and the retrieved mean cloud properties is not universal and while accounting for regional variation is one way to address this, a better understanding of this effect is still needed to reduce uncertainty in aerosol-cloud interactions.

# 1 Introduction

15

The top of atmosphere (TOA) all-sky albedo (α, or, albedo), the fraction of incoming shortwave solar radiation reflected back into space, is a key factor in the planetary energy balance. It governs the difference between absorbed and reflected energy, with the processes that define global albedo playing an essential role in Earth's climate system (Loeb et al., 2007; Trenberth et al., 2009). The present-day estimate of the Earth's mean albedo is 0.29 (Stephens et al., 2015), which aligns with the first satellite-based measurement from the 1970s, where it was determined to be 0.30 (Vonder Haar and Suomi, 1971). Small changes in the albedo can have significant impacts on global mean temperature (Cess, 1976; North et al., 1981), both forcing climate change (Twomey, 1974) and acting as feedbacks to dampen or enhance the climate response to human activity (Budyko, 1969; Hansen et al., 1984). Despite the apparent stability in global mean albedo since the 1970s, satellite records reveal pronounced regional and temporal fluctuations, underscoring the need to understand the cloud processes that shape this balance (Loeb et al., 2024).

The global albedo can be influenced by a wide range of factors (Wielicki et al., 2005), including changes in surface characteristics (Hao et al., 2018, 2019; Miao et al., 2022; Nkemdirim, 1972; Sailor, 1995), as well as atmospheric factors, such

50

55

as aerosol loading and properties (Herman and Browning, 1975), and change to the coverage and properties of clouds (Bender et al., 2011; Engström et al., 2014). With clouds being one of the most important (responsible for approximately half of the Earth's total albedo; Mueller et al., 2011) and variable (Hartmann and Short, 1980) factors, it is vital to understand how changes in cloud properties can modify planetary albedo and hence the overall energy budget.

While the albedo of a field of clouds is known to depend strongly on the cloud coverage and water path, processes that modify these, such as aerosol-cloud interactions (Bellouin et al., 2020) and cloud feedbacks (Stephens, 2005) are expected to have significant impacts on planetary albedo. This has led to several techniques to reconstruct the 'scene' albedo (or changes in it) from cloud properties and their variations. Radiative kernels (Zelinka et al., 2012) link discretised cloud properties to TOA radiative fluxes and have been extensively used to calculate the strength of cloud feedbacks (Ceppi et al., 2016; Zhang et al., 2021) and the impact of aerosols (Wall et al., 2022). With their discretised and almost-linear nature, they are typically applied to monthly-mean cloud and radiative properties. The cloud radiative kernel from Zelinka et al. (2012) uses changes in the mean cloud fraction, optical depth and cloud top pressure to calculate the gridbox TOA shortwave change, although the cloud top pressure has a minimal impact. Kernel methods have been shown to accurately reconstruct cloud feedbacks in model output, comparing well to more complex methods (Zelinka et al., 2012).

Single/multi-variable regressions have also been used to characterise the relationship between cloud properties and albedo (Quaas et al., 2008), forming a critical part of observation-based calculations of the radiative forcing (Bellouin et al., 2020). Some studies have used a single, global relationship (Quaas et al., 2008), whilst others use local regressions, but often only one predictive variable at a time (creating uncertainty in the final result; e.g. Feingold et al. (2017)). CF and LWP have been shown as insufficient for constraining albedo in studies with varying aerosol concentrations (Engström et al., 2015), such that aerosol-cloud-interaction studies typically constrain albedo using the mean CF, LWP and  $N_{\rm d}$ . With relationships between albedo and cloud properties calculated at a climate-model gridbox scale, this method has been shown sufficient for calculating the aerosol forcing in model output (Gryspeerdt et al., 2020).

However, these kernels/relationships assume some linearity between the cloud properties and the albedo. While this can be accounted for by using more bins the radiative kernel (e.g. Gryspeerdt et al., 2019), this does not account for the sub-pixel distribution of cloud properties. For example, with a non-linear relationship between LWP and cloud albedo, the mean LWP of a field of clouds does not uniquely determine the mean cloud albedo (Zhang and Feingold, 2023). It remains unclear how important this effect is at a global scale – to what extent is the scene albedo accurately captured by a three-parameter decomposition (CF, LWP and  $N_d$ )? While climate model results suggest this is sufficient, they do not represent the details of the sub-gridbox cloud distribution that allows this assumption to be accurately tested.

The aim of this study is to assess how well mean cloud properties can be used to reconstruct the albedo of scenes of clouds across the globe. Using a joint-histogram/kernel approach from Gryspeerdt et al. (2019), this work reconstructs albedo from average cloud properties at 100km scales, characterising regional variations in the error in the reconstructed albedo when compared to observations. Different methods for accounting for these biases were assessed, providing recommendations for future observation-based calculations of aerosol forcing and clouds feedbacks.

# 2 Materials and methods

#### 2.1 Data

70

Cloud property retrievals were obtained from the Collection 6.1 MODIS Level-3 (L3) Atmosphere Daily Global Product (MOD08\_D3 and MYD08\_D3 for Terra and Aqua, respectively, Platnick et al., 2017). The Level-3 MODIS Atmosphere Daily Global Product is a daily global spatial aggregation of the parameters generated from the Level-2 MODIS Atmosphere Products: Aerosol (MOD04\_L2, MYD04\_L2), Water Vapor (MOD05\_L2, MYD05\_L2), Cloud (MOD06\_L2, MYD06\_L2), and Atmosphere Profile (MOD07\_L2, MYD07\_L2). The L3 statistics are summarized over a 1 degree equal-angle latitude-longitude grid. The MODIS L3 parameters primarily considered in this study are CF and LWP. Ice cloud fraction (ICF) was additionally used for filtering scenes with overlying ice cloud.

Daily gridded *N*<sub>d</sub> estimates from MODIS were obtained from Gryspeerdt et al. (2022). *N*<sub>d</sub> retrievals in this dataset are based on the Level-2 Collection-6 MODIS Cloud Product (MOD06\_L2, MYD06\_L2) and follow several sampling strategies. This work considers the strategy named G18; proposed by Grosvenor et al. (2018), it balances data quantity with accuracy, accounting for several known biases in the retrieval (Gryspeerdt et al., 2022).

Albedo retrievals were derived from the CERES Daily Time-Interpolated TOA/Surface Fluxes, Clouds, and Aerosols (SSF1deg-Day) product (NASA/LARC/SD/ASDC, 2015b, a) and calculated as the ratio of reflected shortwave radiation to incident solar radiation:

$$\alpha_{\text{CERES}} = \frac{F_{SW}^{TOA}}{F_{FOA}^{TOA}} \tag{1}$$

where  $\alpha_{\text{CERES}}$  represents the observed CERES albedo,  $F_{SW}^{TOA}$  is the observed TOA shortwave flux, and  $F_{solar}^{TOA}$  is the observed TOA solar insolation flux.

Finally, for investigating albedo–cloud sensitivity in stratocumulus region, the ECMWF ERA5 reanalysis (temperature at 700 hPa) was used in order to calculate the estimated inversion strength (EIS), with the formula:

$$EIS = LTS - \Gamma_m^{850} (z_{700} - LCL)$$
 (2)

where LTS is the lower-tropospheric stability,  $\Gamma_m^{850}$  is the moist adiabat at 850 hPa,  $z_{700}$  is the height of p = 700 hPa surface, and LCL is the lifting condensation level. EIS is known to be a predictor of stratus cloud amount (Wood and Bretherton, 2006), and this study assesses the relationship between EIS and differences in albedo estimates to help explaining stratocumulus-to-shallow-cumulus transition.

# 2.2 Method

In order to reduce the overall impact of surface albedo variations, the study area was geographically limited to ocean and latitudes between 60°S and 60°N. All data were filtered through cloud fraction of ice clouds (ICF 

The daily gridded data were binned into discrete intervals based on CF, LWP, and  $N_d$ . Each dimension was divided into a predefined set of ranges, varying from 0 to 1 (linear scale) with 50 bins for CF, from 1 to 1000 gm<sup>-2</sup> (logarithmic scale) with 40 bins for LWP, and from 1 to 300 cm<sup>-3</sup> (logarithmic scale) with 30 bins for  $N_d$ . For each bin, the average albedo ( $\alpha_{avg}$ ) was then calculated as a multi-year mean value of all pixels across the globe that fall into the same bin of CF, LWP, and  $N_d$ .

For each daily gridded MODIS and CERES observation, for each pixel, the difference between the average albedo in the given CF-LWP- $N_d$  bin and the CERES albedo at the given location and time, expressed as  $\Delta \alpha$ , was calculated as:

$$\Delta \alpha = \alpha_{avg} - \alpha_{CERES} \tag{3}$$

The relative difference in estimated albedo was calculated as:

$$\Delta \alpha_{rel} = \left(\frac{\alpha_{avg} - \alpha_{CERES}}{\alpha_{avg}}\right) \cdot 100 \tag{4}$$

Figure 1. Number of days at each location with valid CERES albedo and MODIS cloud properties (CF, LWP and  $N_{\rm d}$ ) across the 2003–2021 years of data included in this study.

# 3 Results

110

115


#### 3.1 Geographic distribution of biases in reconstructed albedo

The percentage of correct albedo estimates depends on the adopted accuracy threshold, which in the present analysis is expressed both in terms of the absolute (Fig. 2a) and the relative (Fig. 2b) value of  $|\Delta\alpha|$ . The results show that when the accuracy threshold is set to  $|\Delta\alpha| = 0.05$ , the reconstruction of albedo is correct in more than 80% of the cases. However, if the stricter threshold of 0.02 is applied, this fraction decreases to 40.91%. Although a deviation of 0.02 (corresponding to 2 percentage points) may be considered relatively small, the analysis presented in panel (b) demonstrates that such a change in absolute terms translates into approximately 10% in relative terms.



Figure 2. Ratio of pixels with correctly estimated albedo as a function of the absolute difference  $|\Delta\alpha|$  (a) and the relative difference  $|\Delta\alpha|$  (b) between the estimated (bin-averaged) and observed CERES albedo. The red dashed lines in panel (a) are the thresholds of  $|\Delta\alpha|$  equal to 0.02, 0.05, and 0.1. In panel (b) they are the relative thresholds of 10, 25 and 50%.

The biases in albedo estimates are not distributed symmetrically around zero. On the contrary, their distribution shows a distinct left-sided asymmetry (Fig. 3). The majority of discrepancies are concentrated around  $\Delta\alpha\approx0.02$ , which corresponds to about 10% relative change. At the same time, cases of underestimate exceeding 50% relative change are clearly observed, whereas such extreme overestimates are practically absent.

Figure 3. Distribution of the absolute difference ( $\Delta \alpha$ ) (a) and the relative difference ( $\Delta \alpha_{rel}$ ) (b) between the estimated (bin-averaged) and observed CERES albedo. The dashed black line marks zero difference, and the red dashed lines indicate the chosen accuracy thresholds ( $\pm 0.02$  for  $\Delta \alpha$ ,  $\pm 10\%$  for  $\Delta \alpha_{rel}$ ).

The spatial distribution of the percentage of underestimated and overestimated albedo values, based on the threshold of 0.02, is illustrated in Figure 4. Both maps reveal well-marked zonal structures. Underestimates of  $\Delta\alpha < -0.02$  are particularly frequent around 40° latitude in both hemispheres, whereas they occur relatively rarely in tropical regions. The opposite tendency is observed for overestimates ( $\Delta\alpha > 0.02$ ). A clear underestimate of albedo is visible over the regions dominated by

marine stratocumulus clouds, with the most distinct example on the west coast of South America highlighted by a red rectangle in both panels of Figure 4. Underestimates are also apparent in mid-latitudes, within regions influenced by the storm tracks of extratropical cyclones. In contrast, overestimate is most commonly observed along the Intertropical Convergence Zone (ITCZ) over the Pacific, Atlantic, and, to some extent, Indian Oceans. In addition to the features of a probable meteorological background, both maps show the presence of some artifacts, which may be related either to the geometry of the observations, such as the solar zenith angle or to the specific features of the retrieval algorithms, as suggested by the faint diagonal lines visible in Figure 4b.

Figure 4. Geographical distribution of the percentage of cases with underestimated (a) and overestimated (b) albedo. Grey colour represents land. The red rectangle marks marine stratocumulus region used in Fig. 6a.

#### 3.2 Drivers of biases in the reconstructed albedo


The zonal patterns shown in Figure 4 become even more apparent when cases of under- and overestimates are separated into individual 5° latitude bands (Fig. 5). On average, the global percentage of underestimates over the analysed multi-year period amounted to 27.7%. This value varies significantly with latitude: from less than 10% around the equator to more than 50% at latitudes higher than 50° in both hemispheres. Similarly, overestimates occurred in 32.1% of cases globally; although the



latitudinal variability was slightly less pronounced than in the case of underestimates, the values still ranged from 15–20% at high latitudes to around 50% in the equatorial zone.

The biases observed over marine stratocumulus regions were examined in greater detail in the context of the transition from thick stratocumulus decks close to the South American coastline towards shallow cumulus clouds occurring further westward over the ocean (the region marked in red in Fig. 4). To investigate this, the relationship between EIS and  $\Delta\alpha$  was analysed for the test period 2007–2019 using Aqua satellite data. Figure 6a presents the joint EIS– $\Delta\alpha$  histogram, normalised by each EIS column. The results show a clear dispersion of  $\Delta\alpha$  values along a characteristic curve: negative values dominate for EIS below 0 K, positive values are typical for EIS between approximately 0 and 15 K, and negative values appear again for EIS above 15 K. EIS values exceeding 15 K most likely then correspond to cases of thick stratocumulus, which in Figure 4a appear predominantly brighter than other cloud scenes with similar CF–LWP– $N_d$  characteristics; clouds with lower EIS are instead associated with shallow cumulus, which often had an overestimated albedo (Fig. 4b). EIS values below 0 K might represent either situations with very limited cloudiness or cases of convective clouds, that is, conditions without a well-defined temperature inversion.

Figure 5. Distribution of (a) under- and (b) overestimated cases of albedo across 5° latitude bands. The boxes show the 25th–75th percentile range, with the line inside marking the median. The whiskers extend to the minimum and maximum values. Blue dots indicate the mean, and black dots mark outliers.





Figure 6b shows the joint histogram of maximum solar zenith angle (SZA<sub>max</sub>) and  $\Delta\alpha$ , normalised by the explanatory variable in each case. The dependence between these two quantities is particularly distinct and appears to explain, to a large extent, the distribution of albedo biases presented in Figure 3. Numerous cases of overestimates around  $\Delta\alpha\approx0.02$  are typical for SZA<sub>max</sub> below around 35°, whereas above ~40° a clear relationship between SZA<sub>max</sub> and  $\Delta\alpha$  exists, with  $\Delta\alpha$  decreasing significantly as SZA<sub>max</sub> increases. For solar zenith angles above 50°, differences in albedo estimates reach values as high as 0.05–0.07. This explains the significant number of strong underestimates also visible in Figure 3.

Figure 6. (a) The joint EIS- $\Delta\alpha$  histogram for the 10°S-30°S, 85°W-65°W region (Aqua only, 2007–2019), marked with a red rectangle on Figure 1; (b) The joint SZA<sub>max</sub>- $\Delta\alpha$  histogram for the global ocean (2003–2021). In both plots each column is normalised so that it sums to 1, showing conditional probabilities  $P(\Delta\alpha|EIS)$ ) (a) and  $P(\Delta\alpha|SZA_{max})$ ) (b). Red dashed lines indicate  $\Delta\alpha=-0.02$  and  $\Delta\alpha=0.02$ , black dashed line –  $\Delta\alpha=0.00$ .

# 3.3 Improving albedo estimates

There is a clear relationship with the geometry of observations, as well as expected differences resulting from the variable properties of clouds. These deviations are not fully captured by the mean cloud properties. In this part of the study several possible methodological adjustments were tested with an aim of improving the accuracy of albedo estimates (Tab. 1).

As a first step, modifications related to cloud fraction were considered. Since it can be expected that for low cloud fraction the varying brightness of the underlying ocean surface (resulting e.g. from the presence of atmospheric aerosol, waves, or the angle of solar rays) may have an larger influence on the albedo, only cases with CF greater than 0.95 were selected for analysis (methodological modification no. I). Secondly, in order to ensure that the number of bins (50) was sufficient to reflect the characteristic U-shaped distribution of cloud fraction (with very small or nearly complete cloud cover occurring most frequently, while intermediate values appear relatively rarely), an alternative estimation was also performed using a much larger number of bins – 1000 (modification no. II).



Two methodological adjustments related to the  $SZA_{max}$  were then tested. In the first case, all pixels with  $SZA_{max} \ge 40^{\circ}$  were excluded in order to eliminate situations where the relationship between  $SZA_{max}$  and  $\Delta\alpha$  becomes approximately linear (modification no. III). The second correction consisted of determining the mean  $\Delta\alpha$  value within each 1°  $SZA_{max}$  bin, and subsequently applying this correction to every estimated albedo case (modification no. IV).

The final group of modifications was aimed at testing possible corrections related to specific local conditions not captured by a global-mean analysis. These included: calculating  $\alpha_{avg}$  separately for each 5° latitude band (modification no. V), calculating  $\alpha_{avg}$  separately for each 5° latitude–longitude grid cell (modification no. VI), and calculating  $\alpha_{avg}$  separately for each 1° latitude–longitude grid cell (modification no. VII). Additionally, a combined adjustment of both filtering out the SZA<sub>max</sub>  $\geq$  40° pixels and computing  $\alpha_{avg}$  separately for 1° grid cell (modification no. VIII) was tested. Table 1 shows the ratio of under- and overestimated cases across the globe for each tested methodological modification.

Table 1. Ratio of underestimated ( $\Delta \alpha < -0.02$ ) and overestimated ( $\Delta \alpha > 0.02$ ) cases of albedo for the considered methodological modifications. Values in brackets indicate the difference with respect to the original method.

| Methodological modification                         | Ratio of cases (%) with: |                        |                          |                            |
|-----------------------------------------------------|--------------------------|------------------------|--------------------------|----------------------------|
|                                                     | $\Delta \alpha < -0.02$  | $\Delta \alpha > 0.02$ | $ \Delta \alpha  > 0.02$ | $ \Delta \alpha  \le 0.02$ |
| Original method                                     | 27.7                     | 32.1                   | 59.8                     | 40.2                       |
| I: CF > 0.95                                        | 24.3 (-3.4)              | 44.3 (+12.2)           | 68.6 (+8.8)              | 31.4 (-8.8)                |
| II: 1000 bins                                       | 27.1 (-0.6)              | 30.3 (-1.8)            | 57.4 (-2.4)              | 42.6 (+2.4)                |
| III: $SZA_{max} < 40^{\circ}$                       | 24.3 (-3.4)              | 17.2 (-14.9)           | 41.5 (-18.3)             | 58.5 (+18.3)               |
| IV: $SZA_{max}$ – $\Delta\alpha$                    | 22.0 (-5.7)              | 24.9 (-7.2)            | 46.9 (-12.9)             | 53.1 (+12.9)               |
| V: 5° latitude                                      | 24.0 (-3.7)              | 26.3 (-5.8)            | 50.3 (-9.5)              | 49.7 (+9.5)                |
| VI: 5° lat, lon                                     | 18.9 (-8.8)              | 19.9 (-12.2)           | 38.8 (-21.0)             | 61.2 (+21.0)               |
| VII: 1° lat, lon                                    | 5.7 (-22.0)              | 5.7 (-26.4)            | 11.4 (-48.4)             | 88.6 (+48.4)               |
| VIII: $SZA_{max} < 40^{\circ} + 1^{\circ}$ lat, lon | 2.7 (-25.0)              | 2.7 (-29.4)            | 5.4 (-54.4)              | 94.6 (+54.4)               |

The most effective improvements in the estimates were achieved when  $\alpha_{avg}$  was calculated separately for individual grid cells. Computing  $\alpha_{avg}$  within 5° latitude bands yielded only a modest reduction in errors, and the total share of incorrect estimates decreased only from 59.8% to 50.3%. At the 1° grid (pixel) level, the error rate dropped nearly sixfold, to 11.4%. Since the kernel was built for the ocean, the underlying surface albedo is most likely not affecting this improvement, and the other sources of variability that weren't considered by the CF–LWP– $N_d$ – $\alpha$  function seem to impact the estimates – possibly other cloud properties and regimes which would be more restricted to location. Both modifications accounting for the SZA<sub>max</sub> – the exclusion from the analysis of pixels with SZA<sub>max</sub>  $\geq$  40°, as well as implementing a correction of mean  $\Delta\alpha$  within each SZA<sub>max</sub> interval – resulted in the improvement that was greater for overestimates than for underestimates. Figure 7a-b shows the histogram of  $\Delta\alpha$  after applying this correction. Many of the corrected biases correspond to overestimates previously identified in the ITCZ region (not shown).



Restricting the analysis to cases with CF > 0.95 did not improve the estimates; instead, it substantially increased the number of overestimates. Increasing the number of CF bins had a very limited effect on the results – the improvement was negligible, showing that 50 bins is already sufficient to represent the non-linearity in the CF-albedo relationship.

The last of the tested modifications (VIII) – a combination of the two methods (IV and VII) – produced the most accurate results: the overall error percentage decreased to 5.4%, i.e. more than tenfold compared to the original value; moreover, the share of under- and overestimates became nearly equal (Fig. 7c-d). Generally, the tests demonstrated that among the investigated modifications, only those accounting for regional and seasonal variability of albedo provided a substantial improvement of the estimates.

Figure 7. Distribution of the absolute difference ( $\Delta \alpha$ ) and the relative difference ( $\Delta \alpha_{rel}$ ) between the estimated (bin-averaged) and observed CERES albedo for modifications no. IV (a-b) and no. VIII (c-d).

#### 4 Discussion






These results show that the reconstructed albedo of a scene of clouds based on the mean cloud field properties exhibits systematic biases, with underestimates prevailing at higher latitudes, in known stratocumulus regions and storm tracks, while overestimates are most pronounced in the tropics and the ITCZ. While these biases are linked to other properties, such as solar zenith angle (Fig. 6b), mean cloud properties are only able to reconstruct the albedo of a scene of clouds accurately when regions are treated separately (Tab. 1).

Regime-dependent retrieval biases could play a role in these biases in the reconstructed albedo. The  $N_d$  retrieval has significant biases in broken cloud fields (Grosvenor et al., 2018), which could contribute to the relationship between  $\Delta\alpha$  and EIS. However, as this  $N_d$  bias exists even in regionally-specific studies, the accuracy of the local reconstructions (Tab. 1) suggests that retrieval biases are unlikely the main factor.

This study focuses on the ocean to reduce the impact of surface albedo variations, but other sources of variation in the clear-sky albedo (such as variations in aerosol loading) might also lead to biases in  $\Delta\alpha$ . The patterns of biases in the  $\Delta\alpha$  field (Fig. 4) are not consistent with the patterns of high aerosol optical depth and the bias remains even when considering only high cloud fraction cases (minimising the impact of surface/clear-sky albedo errors), suggesting that aerosol variability is unlikely to explain these results.

Even limiting this study to the global oceans (restricting surface albedo variability) and accounting for solar zenith angle variations (which corrects for latitude and time of year; Wall et al., 2022), only half of reconstructed albedo values are within 10% of the observed value. This suggests that further properties of the cloud field may be necessary to accurately reconstruct the scene albedo using a single global relationship. The relationship to EIS (Fig. 6a) hints at an importance of cloud regime. The sub-gridbox LWP and cloud optical depth distributions are very different for shallow cumulus (at low EIS) and stratocumulus (at high EIS) (Bretherton et al., 2019). Given the non-linear relationship between LWP and cloud albedo (Platnick and Twomey, 1994), these regional variations could create local biases in the reconstructed albedo. If this additional parameter changes in response to aerosol loading, such as during the transition between open and closed celled stratocumulus (Rosenfeld et al., 2006), this could lead to an additional component of the aerosol forcing not covered by the usual decomposition into Twomey (*N*<sub>d</sub>), LWP and CF adjustments common in observation-based studies (Bellouin et al., 2020). As this effect is not represented in climate models, the closure achieved in previous studies (e.g. Gryspeerdt et al., 2020) could be misleading.

The high accuracy/low Δα achieved when using locally-specific kernels identifies a near-term pathway to address this issue, demonstrating that with the correct explanatory variables, an accurate albedo reconstruction is possible. However, it also highlights the need to identify these additional factors that control the albedo of a cloud scene, beyond the gridbox mean cloud properties.

# **5 Conclusions**






The two dominant factors determining liquid marine clouds albedo ( $\alpha$ ) are cloud fraction (CF) and liquid water path (LWP), with aerosol loading further impacting albedo's variability, giving additional benefits from using  $N_d$  as a further explanatory variable (Bellouin et al., 2020; Engström et al., 2015).

This study uses near-global (60°S–60°N) MODIS and CERES retrievals for marine regions from Terra and Aqua satellites spanning over the years 2003–2021 to build a simplified CF–LWP– $N_d$ – $\alpha$  kernel and examine the spatial differences in albedo-to-cloud-sensitivity.

It was demonstrated that the number of biases in reconstructed albedo can be as high as ~60% of cases, when aiming for the accuracy in estimates (absolute difference between expected for the given CF–LWP–N<sub>d</sub> conditions and measured by CERES albedo) at 0.02; which corresponds to about 10% of relative difference (Fig. 3). Moreover, it was showed that the CF–LWP–N<sub>d</sub>–α relationship varies significantly across the globe. Underestimates are particularly frequent in regions of known stratocumulus regimes – on the west coast of South America, California and South Africa – as well as along the midlatitude storm tracks. In contrast, overestimates occur mainly in the tropics. Clear zonal patterns in the accuracy of albedo estimates suggested a potential relationship to the solar zenith angle of satellite observations. The average Δα for SZA<sub>max</sub> up to 30–35° was found to be largely constant at a 0.02 overestimate. For the SZA<sub>max</sub> > ~40° however, Δα decreases significantly with the increase of SZA<sub>max</sub>, reaching about –0.05 for SZA<sub>max</sub> higher than 60°.

Several modifications of the initial method were tested in attempt to improve the albedo estimates. The highest number of correct estimates ( $\sim$ 94.6% for  $|\Delta\alpha| \le 0.02$ ) can be achieved when the average albedo in the given CF–LWP– $N_d$  conditions is calculated at a 1° grid resolution. Generally, correcting for SZA<sub>max</sub> significantly decreases the number of overestimates with modest decrease in underestimates.

Although it was showed that there are some geographical patterns in CF–LWP– $N_d$ – $\alpha$  relationship and possible modifications were suggested to improve the albedo estimates, significant uncertainties remain. Performing the estimates on a pixel level and thus, correcting for specific local conditions, significantly reduces the biases, suggesting that there are explanatory variables that could be used, beyond the mean cloud field properties. However, the factors driving this regional variation are not yet clear, and a wider study further addressing this will be essential for a more complete understanding.

**Code availability:** All plots and calculations were produced with custom Python code. The code can be obtained by contacting the corresponding author.

**Data availability:** The MODIS Level-3 Atmosphere Daily Global Product (MOD08\_D3, MYD08\_D3) was obtained through the Level-1 and Atmosphere Archive & Distribution System (LAADS) Distributed Active Archive Center (DAAC) (https://doi.org/10.5067/MODIS/MOD08\_D3.061 and https://doi.org/10.5067/MODIS/MYD08\_D3.061, Platnick et al., 2017). The CERES Daily Time-Interpolated TOA/Surface Fluxes, Clouds, and Aerosols (CER\_SSF1deg-Day) product was downloaded from NASA's Langley Research Center via https://ceres-tool.larc.nasa.gov/ord-

tool/jsp/SSF1degEd41Selection.jsp (https://doi.org/10.5067/TERRA/CERES/SSF1DEGDAY\_L3.004 and https://doi.org/10.5067/AQUA/CERES/SSF1DEGDAY\_L3.004A, NASA/LARC/SD/ASDC, 2015b, a). The gridded cloud droplet number concentration ( $N_d$ ) dataset is available at the Centre for Environmental Data Analysis (CEDA) at https://doi.org/10.5285/864a46cc65054008857ee5bb772a2a2b (Gryspeerdt et al., 2022).

Author contribution: Both authors contributed to the study design and were involved with the interpretation of results. IW performed the analysis and prepared the manuscript with comments from EG.

**Competing interests:** The authors declare that they have no conflict of interest.

**Financial support:** Izabela Wojciechowska was supported by a short-term research grant from the University of Warsaw, and the research was carried out at Imperial College London. The study was funded by the University of Warsaw. Edward Gryspeerdt was supported by a Royal Society University Research Fellowship (URF/R1/191602).

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
