# Peer review of "Reconstructing albedo from mean cloud properties"

_EGUsphere, 2025_

## Referee Comment (RC1)

**Review of Wojciechowska et al., 2025.**

This paper tests the performance of a simplified calculation of all-sky albedo vs CERES satellite data. The simplified calculation consists of a function or look-up table (LUT) using the MODIS-observed 1x1 degree daily cloud fraction (CF), droplet number (Nd) and Liquid Water Path (LWP) values as inputs. It is constructed using a kernel approach that (presumably – there needs to be more detail on this in the paper) uses several other MODIS variables as inputs along with the CF, Nd and LWP. Using a single (time and global) mean LUT leads to large errors that exhibit a spatial pattern and a dependence on Estimated Inversion Strength (EIS) and Solar Zenith Angle (SZA). Attempts were made to improve the LUT. Correcting for the SZA bias using the single global mean LUT had only a small impact. However, moving to using a separate time-averaged LUTs for each 1x1 gridbox led to significant improvements leaving only small errors. This suggests that regional information (in addition to the CF, Nd and LWP values) is needed for an accurate estimate of albedo.

The paper describes a potentially very useful simplified way to calculate albedo quickly and easily based on only 3 cloud variables. I recommend its publication after the changes below are made – mostly clarifications of the methods and some extra description.

**General and line-by-line comments**

It is interesting that there is no need for separate seasonal/time-varying LUTs – one time-mean LUT for the whole data period for each location seems sufficient to get low errors. It might be worth commenting on this a little more.

Is there an advantage to using the LUTs vs using the kernel method directly? Especially if there is a need to have a separate LUT for each grid-box requiring a fairly large array to be stored? If we knew what is required by the kernel method then this might be clearer to the reader – presumably it requires lots of extra variables (see comments later)?

Line 94 - How much does the filtering by ice cloud fraction restrict the altitude of the clouds studied? It would be useful to show this somewhere. E.g., are you just looking at low-altitude clouds after the filtering?

Line 96 – it would be useful to reiterate that this is 1x1 degree data. E.g., "The daily gridded 1x1 degree data...".

Line 98 – "For each bin, the average albedo (  $\alpha$  avg) was then calculated as a multi-year mean value of all pixels across the globe that fall into the same bin of CF, LWP, and Nd."

- It's not clear from the methods section how you calculate the albedo of the pixel using CF, LWP and Nd. Presumably, it is as mentioned on line 62 ("Using a joint-histogram/kernel approach from Gryspeerdt et al. (2019),")? But this should be described in the methods section too. Some details on how the method works should be provided too.
- Otherwise one might think that you could use the bin-centre values to calculate albedos for each bin using the kernel method without having to do it for every datapoint and then averaging? But I think this comes from the lack of explanation

- about the kernel method. Presumably the kernel method requires more information so that this is not possible? It would be good to talk about that a little.
- It would also be good to say that the average albedo values for each bin are the ones that could form the "look-up table" that might enable rapid albedo calculations based on just CF, Nd and LWP, which would be a lot easier than doing radiative calculations and (presumably) easier than doing the kernel calculation. And then that this approach needs to be tested against CERES (following onto the next sentence). This would be useful to the reader because it is reiterating the aim of the paper, but at the point in the text where you have explained the approach.
- The word "pixel" here is a bit confusing too "1x1 degree daily datapoints" would be clearer.

Line 132: "Underestimates of  $\Delta$   $\alpha$  < -0.02 are particularly frequent around 40 $^{\circ}$  latitude in both hemispheres,"

It looks to me like the frequencies are high at latitudes greater than 40 deg?

Line 140: "as suggested by the faint diagonal lines visible in Figure 4b."

- I can't really see any faint diagonal lines? I can see some straight lines that look like artefacts, though.

Line 160: "which in Figure 4a appear predominantly brighter than other cloud scenes with similar CF-LWP-Nd characteristics"

 Could be worded better. Fig. 4a more suggests that they "are observed by CERES to be brighter than calculated from the CF-LWP-Nd values using the kernel approach", or similar.

Line 175: "This explains the significant number of strong underestimates also visible in Figure 3."

- It might also suggest why there are underestimates at high latitudes in Fig. 4a.

Line 190: "Secondly, in order to ensure that the number of bins (50) was sufficient to reflect the characteristic U-shaped distribution of cloud fraction (with very small or nearly complete cloud cover occurring most frequently, while intermediate values appear relatively rarely), an alternative estimation was also performed using a much larger number of bins - 1000 (modification no. II)."

- Presumably, this is separate to modification no. I? It would be good to make that clear here.

Line 216 – "Figure 7a-b shows the histogram of  $\Delta\alpha$  after applying this correction." – it's not clear which correction you are referring to here. From the text and table I think that this is just the correction using the mean  $\Delta\alpha$  within each SZAmax interval (modification IV) and not also modification III?

Line 232 (and 235) – "These results show that the reconstructed albedo of a scene of clouds based on the mean cloud field properties exhibits systematic biases"

- "based on the mean cloud field properties" is a bit confusing here since you are basing it on the actual CF, LWP and Nd cloud properties the issue rather seems to be that using a single global mean "look-up table" with mean albedo values for each bin leads to systematic biases?
- Reading on to line 260 makes it clearer what you mean here since you have now explained that there are likely factors other than the cloud properties (CF, Nd and LWP) at play. However, this was not so clear at the start of the section where you should explain the use of the global mean albedo look-up-table (as mentioned in the previous bullet point) and mention that by "mean cloud properties" you mean CF, Nd and LWP only.

Line 268 – "to build a simplified CF-LWP-Nd-  $\alpha$  kernel" – not sure if this is a very descriptive way of describing it. "a simplified method to calculate albedo based only on CF, LWP and Nd values" or similar would be better.

Line 268 – "spatial differences in albedo-to-cloud-sensitivity.". This is also not clear – do you mean "spatial differences in the sensitivity of albedo to cloud properties"?

Line 270 – "It was demonstrated that the number of biases in reconstructed albedo can be as high as ~60% of cases, when aiming for the accuracy in estimates (absolute difference between expected for the given CF-LWP-Nd conditions and measured by CERES albedo) at 0.02; which corresponds to about 10% of relative difference (Fig. 3)."

- This would be better as "It was demonstrated that the percentage of datapoints in which the reconstructed albedo biases (relative to the measured CERES albedo) were >+/-0.02 (a relative bias of around +/-10%) can be as high as ~60% (Fig. 3)."

Line 280 "can be achieved when the average albedo in the given CF-LWP-Nd conditions is calculated at a 1 $^{\circ}$  grid resolution."

- Would be better as "can be achieved when the average albedo for each CF, LWP and Nd bin is calculated at a 1° grid resolution."

Line 284: "on a pixel level" – again, better as "at a 1 degree resolution"

Line 286 – "the mean cloud field properties" – again, it would be good to say that you mean CF, Nd and LWP here.

**Typos**

Line 55; "bins the" -> "bins in the"

Line 90:" explaining" -> "explain"

Line 95: "Resulting subset of cases considered in this study is pictured at Figure 1." -> "The resulting subset of cases considered in this study is pictured in Figure 1."

Line 189: "may have an larger influence" -> "may have a larger influence"

Line 272: "showed" -> "shown".

Line 279: "in attempt" -> "in an attempt"

Line 282: "with modest" -> "with a modest"

Line 283: "showed" -> "shown"

Line 283: "in CF-LWP-Nd-  $\alpha$  " -> "in the CF-LWP-Nd-  $\alpha$  "

---

## Referee Comment (RC3)

Wojciechowska and Gryspeerdt present a simplified framework for estimating marine liquid cloud albedo using cloud fraction (CF), liquid water path (LWP), and cloud droplet number concentration (Nd). Their approach combines climatological, collocated MODIS cloud properties with CERES albedo to construct a reduced cloud albedo kernel. The authors show that this formulation yields robust albedo estimates, with more than 80 percent of samples differing from CERES by less than 0.05. The results also exhibit a clear spatial structure in regions where albedo is systematically underestimated or overestimated.

The manuscript is clearly written, and I find no major issues with the methodology or the interpretation of results. My comments concern a few missing details and clarifications. I also suggest a straightforward way to incorporate cloud morphology into the algorithm, which may further enhance the albedo estimates. I recommend acceptance of the manuscript after the following minor comments are addressed:

Line 57: One of the key factors that controls the relationship between albedo and CF and/or LWP and/or Nd/CER is the cloud morphology or sub-grid cloud heterogeneity (Goren et al., 2023; McCoy et al., 2023; Choudhury and Goren, 2024).

It is still unclear to me what the main motivation is for assessing how accurately cloud albedo can be estimated from cloud microphysical and macrophysical properties. Is the aim to evaluate these relationships because they are used to quantify changes in cloud albedo driven by changes in cloud properties, as in Zhang et al. (2021) or Wall et al. (2022)? If so, then this study seems to assess whether kernel-based decompositions of cloud albedo are justified. Could the authors clarify this point in the introduction?

Line 73: Which cloud fraction is used? Is it the cloud fraction derived only from pixels with successful cloud property retrievals, as is common in many aerosol–cloud interaction studies, or the total cloud fraction including partially cloudy pixels and those with failed retrievals?

Line 79: Were CERES Aqua and Terra products considered separately and paired with their respective MODIS-cloud retrievals, or were the datasets averaged across the two platforms?

Section 3.1: I recommend including an additional global map showing the geographical distribution of albedo bias (potentially as an extension of Figure 4). This figure should be supported by a short description summarizing the mean bias and its spatial pattern, noting regions where the discrepancies are especially pronounced.

Section 3.1: Does the bias decrease when the data are resampled to the monthly scale, which is the temporal resolution typically used in studies that construct such kernels?

Figure 6b: How does the solar zenith angle extend beyond 65 degrees when this threshold was used in the G18 sampling strategy for estimating Nd?

Line 213: "... more restricted to location ..." Perhaps "cloud morphology" is intended here?

Line 224: What explains the seasonal variability? Did you construct the kernel at each grid point as well as for all days of the year separately? If not, this might help reduce the bias further.

I also suggest adding thin cloud fraction (F\_thin, a proxy for cloud morphology) to the kernel. F\_thin can be computed from the MODIS Level-3

"Cloud\_Optical\_Thickness\_Liquid\_Histogram\_Counts" dataset by using COT < 3 or < 5. Including F\_thin alongside CF, LWP and Nd/CER may improve the albedo estimation (McCoy et al., 2023).

Could the authors speculate on why albedo is underestimated in stratocumulus regions and overestimated in the tropics?

**Language edits:**

Line 55: "more bins in ..."?

Line 55: "sub-pixel" or "sub-gridbox"

Line 101: By "pixel," do you mean the 100 km gridbox?

Line 123: Do you mean "left-skewed"?

Line 270: Please rephrase. The current wording is unclear.

**References:**

Choudhury, G., & Goren, T. (2024). Thin clouds control the cloud radiative effect along the Sc-Cu transition. *Journal of Geophysical Research: Atmospheres*, 129, e2023JD040406. <a href="https://doi.org/10.1029/2023JD040406">https://doi.org/10.1029/2023JD040406</a>

Goren, T., Sourdeval, O., Kretzschmar, J., & Quaas, J. (2023). Spatial aggregation of satellite observations leads to an overestimation of the radiative forcing due to aerosol-cloud interactions. *Geophysical Research Letters*, 50, e2023GL105282. https://doi.org/10.1029/2023GL105282

McCoy, I. L., McCoy, D. T., Wood, R., Zuidema, P., & Bender, F. A.-M. (2023). The role of mesoscale cloud morphology in the shortwave cloud feedback. *Geophysical Research Letters*, 50, e2022GL101042. <a href="https://doi.org/10.1029/2022GL101042">https://doi.org/10.1029/2022GL101042</a>

Wall, C. J., Norris, J. R., Possner, A., McCoy, D. T., McCoy, I. L., and Lutsko, N. J.: Assessing effective radiative forcing from aerosol-cloud interactions over the global ocean, Proc. Natl. Acad. Sci. U. S. A., 119, 1–9, 2022.

Zhang, Y., Jin, Z., and Sikand, M.: The Top-of-Atmosphere, Surface and Atmospheric Cloud Radiative Kernels Based on ISCCP-H Datasets: Method and Evaluation, J. Geophys. Res. Atmos., 126, e2021JD035053, <a href="https://doi.org/https://doi.org/10.1029/2021JD035053">https://doi.org/https://doi.org/10.1029/2021JD035053</a>, 2021.